# Tissue Lipid Profiles of Rainbow Trout, *Oncorhynchus mykiss*, Cultivated under Environmental Variables on a Diet Supplemented with Dihydroquercetin and Arabinogalactan

**DOI:** 10.3390/ani14010094

**Published:** 2023-12-27

**Authors:** Natalia N. Fokina, Irina V. Sukhovskaya, Nadezhda P. Kantserova, Liudmila A. Lysenko

**Affiliations:** Laboratory of Environmental Biochemistry, Institute of Biology, Karelian Research Centre of the Russian Academy of Sciences, 185910 Petrozavodsk, Russia; fokinann@gmail.com (N.N.F.); sukhovskaya@inbox.ru (I.V.S.); l-lysenko@yandex.ru (L.A.L.)

**Keywords:** rainbow trout, dietary supplement, temperature response, lipid store, fillet composition

## Abstract

**Simple Summary:**

Plant-origin supplements are safe dietary ingredients that improve the growth, digestion and stress tolerance of farmed fish and also contribute to the safety of natural ecosystems and the sustainability of aquaculture. These compounds appear to accelerate the tolerance of fish to environmental stressors, which are numerous in fish production, and improve the quality of fish products, a good source of dietary protein and lipids in the human diet. A four-month feeding study of a diet supplemented with a mixture of the antioxidant dihydroquercetin and the probiotic arabinogalactan was designed to determine possible effects on the lipid profile and nutritional value of rainbow trout reared in cages under natural environmental variables. When none of the environmental factors disturbed the growth of the trout, triacylglycerols with a lower unsaturation index of the fatty acid moiety were mainly stored in both the liver and muscle, irrespective of the dietary supplement. However, fish fed the supplement were more tolerant to the summer temperature rise, demonstrating a lower mortality rate, better performance, stable lipid accumulation and a higher ratio of n-3 to n-6 unsaturated fatty acids. Thus, when used prophylactically, this dietary supplement is beneficial, both for the tolerance of fish to environmental stressors and for the lipid composition of fish fillets intended for human consumption.

**Abstract:**

Reared rainbow trout are vulnerable to environmental stressors, in particular seasonal water warming, which affects fish welfare and growth and induces a temperature response, which involves modifications in tissue lipid profiles. Dietary supplements of plant origin, including the studied mix of a flavonoid, dihydroquercetin and a polysaccharide, arabinogalactan (25 and 50 mg per 1 kg of feed, respectively), extracted from larch wood waste, were shown to facilitate stress tolerance in fish and also to be beneficial for the safety of natural ecosystems and the sustainability of aquaculture production. This four-month feeding trial aimed to determine the effects of the supplement on liver and muscle lipid accumulation and the composition in rainbow trout reared under environmental variables. During periods of environmental optimum for trout, a consistent increase in energy lipid stores, particularly triacylglycerols (2.18 vs. 1.49-fold over a growing season), and an overall increase in lipid saturation due to lower levels of PUFAs, such as eicosapentaenoic (20:5n-3), docosahexaenoic (22:6n-3) and arachidonic (20:4n-6) acids, were observed in both control and supplement-fed fish, respectively. However, in fish stressed by an increase in ambient temperature, dietary supplementation with dihydroquercetin and arabinogalactan reduced mortality (3.65 in control vs. 2.88% in supplement-fed fish, *p* < 0.05) and alleviated the high-temperature-induced inhibition of lipid accumulation. It also stabilised the membrane phospholipid ratio and moderated the fatty acid composition of fish muscle and liver, resulting in higher levels of n-3 PUFAs and their precursors. Thus, the natural compounds tested are beneficial in accelerating fish tolerance to environmental stressors, reducing mortality and thermal response, and moderately improving fillet quality attributes by increasing the protein/lipid ratio and the abundance of fatty acids essential for human nutrition.

## 1. Introduction

In fish aquaculture production, various approaches are used to increase the efficiency of fish farming and improve the quality of the final commercial products. The nutrient composition of formulated fish feeds, in particular, a balance and a source of fats and proteins, undoubtedly has the greatest impact on the welfare, growth rate and tissue composition of fish. Dietary fats are a source of energy, essential fatty acids and fat-soluble vitamins for fish, but their excessive supply with a high-fat diet can limit the assimilation of essential nutrients, thereby reducing the growth rate and enhancing body fat deposition in farmed fish. Conversely, the depletion of intramuscular fat and protein in fish due to malnutrition or ambient stressors, such as hyper- or hypothermia, infectious diseases, etc., led to deterioration in muscle quality [1,2]. In addition to the total lipid accumulation, ambient temperature affects the fatty acid composition of the membrane phospholipids to maintain cell membrane permeability in a conservative manner that has been studied in detail [3,4,5]. The quantity and quality of intramuscular lipids are important for the texture, flavour and nutritional value of the fillet. Dietary supplementation with biologically active compounds of plant origin could be a way to improve the health status of farmed fish under intensive culture and ambient stressors and to modify the nutritional quality of the fillet [6,7,8,9]. However, little is known about the effects of natural dietary supplements on lipid-mediated physiological responses, such as thermal response, and the lipid composition of supplement fed fish [10,11,12]. In our feeding trial, we used dihydroquercetin, a flavonoid isolated from Siberian and Daurian larch wood, which may be beneficial in fish production due to its antioxidant, anti-inflammatory and antimicrobial activities [6,13,14,15,16,17], in a combination with arabinogalactan, a plant polysaccharide of the same origin, which has immunomodulatory effects [18,19,20,21]. 

The pharmacological effects and mechanisms of action of dihydroquercetin (or taxifolin) have recently been reviewed [22,23,24]. Dihydroquercetin is a potent antioxidant and, thus, directly reduces the deleterious effects of ROS through radical scavenging. The enzymatic antioxidant activity of cell homogenates increases in response to low doses of dihydroquercetin, although it appears to decrease at higher doses [6,25]. Dihydroquercetin also inhibits the activity of pro-oxidative enzymes, such as myeloperoxidase, NADPH oxidase, cyclooxygenase and nitric oxide synthase [26]. As a flavonoid, dihydroquercetin has been shown to have anti-inflammatory properties [22,23,24], reducing the production of pro-inflammatory cytokines such as interleukin 1β (IL-1β) and TNF-α [27,28]. Dihydroquercetin affects lipid synthesis, cholesterol production and the ApoA/ApoB balance, although these effects are not caused by changes in transcription levels [29,30]. In cell and animal models, arabinogalactan is able to enhance natural killer cells and macrophages, as well as the secretion of pro-inflammatory cytokines. Arabinogalactan may act indirectly through microbiota-dependent mechanisms and/or have a direct effect on the immune system via gut-associated lymphoid tissue [31]. Although both compounds have different activities, we can assume that they have synergistic or mutual potential [32,33].

This study was conducted to elucidate the effects of a dietary supplement containing natural substances, dihydroquercetin and arabinogalactan, on lipid accumulation and the fatty acid composition in the liver and muscle of rainbow trout, *Oncorhynchus mykiss*, reared under environmental variables.

## 2. Materials and Methods

### 2.1. Experimental Diet Preparation

The basal diet for rainbow trout was obtained commercially (BioMar, Brande, Denmark) and formulated to contain 23–24% crude lipids and 38% crude protein. The analysed lipid composition of the two used basal diet fractions is detailed in Table 1. The basal diet without any supplements was maintained as the control diet. The experimental diet consisted of the basal diet supplemented with 25 mg/kg of dihydroquercetin and 50 mg/kg of arabinogalactan in accordance with the recommendations of the manufacturer (https://dkv99.ru/syrye/digidrokvertsetin-v-zhivotnovodstve/, accessed on 4 December 2023; Quality and Safety Certificate no. 396-08.17, Ametis, Blagoveshchensk, Russia) and the efficacy supported by our previous work [33,34]. The supplements (dihydroquercetin and arabinogalactan) were top-dressed as feed granules by the staff of the farm directly on the day of feeding. For this, a portion of the supplement was dissolved in water at 50 °C in a 10 L tank, and the solution was sprayed onto feed pellets while mixing the feed manually.

### 2.2. Experimental Design and Fish Sampling 

During the in-growing season from May to September, rainbow trout *Oncorhynchus mykiss* (Walbaum) (sexually immature, age 2+) were grown in cages (size 10 × 10 m, area of 100 m^2^, volume of 900 m^3^, distanced from each other on 70 m) on a fish farm (Ladmozero Lake, Russia). The fish were fed one of two diets (in duplicate), either the basal diet BioMar (Denmark) without any supplements (control diet) or the basal diet supplemented with 25 mg/kg of dihydroquercetin and 50 mg/kg of arabinogalactan (experimental diet). The feed was delivered to the fish manually twice a day: 6 mm fraction from May to June and 8 mm fraction from July to September, according to the average weight of fish. Fish in experimental group received the supplemented diet for two-week time slots, alternating with two-week periods on the standard diet. Feeding the supplement in short, repeated courses made it possible to achieve the desired biological effect at reduced economic cost.

Here, we describe the results of the second year of observation on trout grown with the same feeding protocol since age 1+; the results of the first year of the feeding trial were detailed earlier [34]. The average-sized individuals of both groups (733.0 ± 58.3 g for control and 770.1 ± 56.5 g for experimental) were included in the present study. No signs of infectious or parasitic diseases were observed in fish during growing trial. The water temperature and dissolved oxygen content were measured daily (Figure 1). To prevent fish lethality, during the period of water temperature maximum (from 26 July to 5 August), trout were maintained without food. Each month, on 12 May, 27 June, 17 July, 24 August and 18 September, eight fish were selected at random per cage, slaughtered with a blow to the head and cutting the vertebra with a lancet and individually weighed and measured. Individual white muscle and liver samples were fixed in 97% ethanol and stored at +4 °C.

### 2.3. Biometrical Measurements and Survival 

At the end of the feeding trial, all remaining fish in each cage were counted and bulk-weighed for calculation of survival, growth and feed utilization parameters. During the trial, the initial and final weights of fish in each sample were measured individually using a portable Ohaus Scout SPX Series balance. Weight data were used to calculate average weight, relative growth rate (RGR), specific growth rate (SGR) and feed conversion ratio (FCR) per cage for each diet (control and experimental), according to the following equations: RGR (%) = 100 × [(final weight − initial weight)/initial weight](1)
SGR (% day^−1^) = 100 × [(ln final weight − ln initial weight)/days](2)
FCR = total dry feed fed (g)/total weight gain (g)(3)

### 2.4. Analyses of Tissue Lipids 

Biochemical assays for total lipid contents (phospholipids, triacylglycerols, sterols and sterol esters) in *O. mykiss* muscle and liver were conducted in the Equipment Sharing Centre of the Karelian Research Centre, Russian Academy of Sciences (Petrozavodsk, Russia). Total lipids were extracted from fish tissues in chloroform/methanol (2:1, by vol) according to Folch et al. [35]. Extracted lipids were spotted onto silica gel thin-layer chromatography plates (TLC Silica gel 60 F254 plates, Merck, Darmstadt, Germany) and separated into different fractions of lipid classes using petroleum ether/diethyl ether/acetic acid (90:10:1, *v*/*v*) as mobile phase. The position of the fractions was determined by standards: phospholipid mixture (Supelco, St. Louis, MO, USA), cholesterol (Sigma-Aldrich Co., St. Louis, MO, USA), glyceryl trioleate (Sigma-Aldrich Co., St. Louis, MO, USA) and cholesteryl palmitate (Sigma-Aldrich Co., St. Louis, MO, USA). Using the calibration curves for the standards, the ester-bond-containing lipid classes, such as phospholipids, triacylglycerols and sterol esters, were calculated. The quantitative composition of sterol fraction was measured usingthe Engelbrecht et al. [36] method using UV/Vis spectrophotometer (SF-2000, Saint Petersburg, Russia). The total lipid content was expressed as weight % of tissue dry weight, as well as lipid class content, expressed as weight % of total lipid dry weight. The dry weight of tissue was the weight of tissue remaining after extraction plus total lipid dry weight.

The composition of includendividual phospholipid fractions, includeding phosphatidylinositol (PI), phosphatidylserine (PS), phosphatidylethanolamine (PEA), phosphatidylcholine (PC), lysophosphatidylcholine (LPC) and sphingomyelin (SM), was determined by high-performance liquid chromatography (Aquilon Stayer chromatograph, Moscow, Russia) using Arduini et al.’s [37] technique on a Nucleosil 100-7 column (Elsiko, Moscow, Russia) with the liquid phase acetonitrile/hexane/methanol/phosphorus acid (918:30:30:17.5, *v*/*v*) and UV-spectrophotometer with 206 nm wavelength. Peaks were identified by reference to retention times of authentic standards, including phospholipid mixture (P3817, Supelco, St. Louis, MO, USA), phosphatidylserine (Sigma-Aldrich Co., St. Louis, MO, USA) and sphingomyelin (Sigma-Aldrich Co., St. Louis, MO, USA).

### 2.5. Analysis of Fatty Acids 

Fatty acid methyl esters (FAMEs) were obtained via esterification of total lipids using methanol and acetyl chloride, then separated and quantified using gas–liquid chromatography (Agilent 7890A, Agilent Technologies, Santa Clara, CA, USA) equipped with a flame ionization detector, a column DB-23 (60 m × 0.25 mm i.d.) and a split injection system with nitrogen as a carrier gas. FAMEs were identified by comparing retention time with those of authentic fatty acid standards (Supelco, St. Louis, MO, USA).

### 2.6. Statistical Analyses 

Data obtained were expressed as means and standard deviation of means (±SD). Statistical significance was determined by one-way analysis of variance (ANOVA), followed by a post hoc Tukey HSD test on StatSoftStatistica v 7.0 software computer program. The correlations of the studied parameters were analysed using the Pearson coefficient. Multivariate factor analysis (varimax method) was used to analyse the structure of correlations (marked loading are >0.7). For the remaining analyses, the significance level was set at 0.05.

## 3. Results

### 3.1. Growth Performance and Mortality

Table 2 shows the effect of a dietary dihydroquercetin and arabinogalactan mix on the growth parameters and mortality of *O. mykiss*. The average weight of both control and experimental fish was similar at the beginning (12 May) and at the end (18 September) of the observations. By the end of the feeding trial, significant differences were found in terms of whole-season mortality (3.65 in control vs. 2.88% in supplement-fed fish, *p* < 0.05), while no effects on growth performance, specific growth rate and feed utilization were revealed between the *O. mykiss* fed the two diets.

### 3.2. Tissue Lipid Composition

The tissue lipid content and their main fraction composition in supplement-free and supplement-fed fish were mostly similar at the beginning of the observations and substantially differentiated at two sampling points, in July and September. In July, in environments optimal for growth, muscle lipid accumulation prevailed in fish reared on a standard diet; particularly, it accumulated (if compared with supplement-fed fish) total lipids 32.45 ± 3.59 vs. 22.59 ± 2.58, phospholipids 8.64 ± 0.60 vs. 6.09 ± 0.44, cholesterol (Chol) 2.74 ± 0.37 vs. 1.93 ± 0.19, phosphatidylserine (PS) 0.17 ± 0.01 vs. 0.12 ± 0.01, phosphatidylethanolamine (PEA) 1.48 ± 0.11 vs. 1.03 ± 0.08, phosphatidylcholine (PC) 6.66 ± 0.46 vs. 4.70 ± 0.34 and phosphatidylinositol (PI) 0.31 ± 0.02 vs. 0.22 ± 0.02 (% of dry weight, *p* < 0.05). By September, the muscle of control trout had been partially lost in main lipid fractions, while supplement-fed fish become prevalent in total lipids 22.18 ± 1.51 vs. 36.15 ± 5.55, phospholipids 7.20 ± 0.26 vs. 8.76 ± 0.96, and triacylglycerols (TAGs) 10.46 ± 1.04 vs. 21.04 ± 3.61 (given as % of dry weight, *p* < 0.05). The muscle of standard-fed trout was also insufficient in phosphatidylcholine (PC) 5.49 ± 0.20 vs. 6.85 ± 0.84 and cholesterol ethers 2.09 ± 0.15 vs. 3.03 ± 0.48 (control vs. experimental diet, *p* < 0.05) by the end of the season. Hepatic lipid reserves progressively rose from May to September due to main lipid fractions, including total lipids, TAGs, phospholipids, Chol, cholesterol ethers and PC, with their prevalence in the standard-fed fish in July and without a statistical difference (except for PS and PEA/PC ratio) between the diets by the end of the experiment (Figure 2 and Figure 3). Thus, in supplement-fed trout (if compared with the control), the throughout-season total lipid accumulation in white muscle and liver was more consistent and less perturbed by the mid-summer temperature peak.

The membrane phospholipid ratio, PEA to PC (Figure 3), varied with season, being maximal in fish sampled in May (at the lowest ambient temperatures), particularly in supplement-fed fish (receiving the supplement in 2017; see Experimental Design and Fish Sampling).

### 3.3. Tissue Fatty Acid Composition

The fatty acid composition of total lipids extracted from both the muscle and liver of fish showed a seasonal increase in the desaturation rate estimated by the relative content of monounsaturated to saturated fatty acids (MUFAs/SFAs), including palmitoleic to palmitic (16:1n-7/16:0) and oleic to stearic (18:1n-9/18:0). From May to September, the ratio MUFA/SFA increased mostly in a diet-independent manner while being significantly higher in the muscles of fish fed the supplemented diet by the end of the observation. Due to a significant (on 25–50%, *p* < 0.05) decrease in total fatty acid unsaturation and particularly in the content of PUFAs, including arachidonic (ArA; 20:4n-6), eicosapentaenoic (EPA; 20:5n-3) and docosahexaenoic (DHA; 22:6n-3) acids, in the organs of growing fish (from May to September; Figure 4), the SFA/PUFA ratio increased within the season similarly in both fish groups (Figure 5). A significant diet-related difference was noted in the liver sampled in July; the total unsaturation index in supplement-fed fish exceeded those in the control (9.62 ± 0.52 vs. 7.92 ± 0.46, *p* < 0.05) due to an elevated level of total n-6 PUFA (11.72 ± 0.35 vs. 10.40 ± 0.32, *p* < 0.05). The muscle fatty acid composition resembled diet-related differences from June/July, with a higher ratio of n-3 to n-6 fatty acids (1.05 ± 0.08 vs. 0.91 ± 0.05, *p* < 0.05), while a lower unsaturation index (8.73 ± 0.13 vs. 9.47 ± 0.21, *p* < 0.05) and higher ratio of SFA to PUFA (0.57 ± 0.01 vs. 0.53 ± 0.01, *p* < 0.05) in supplement-fed fish was found compared to in the control (Figure 5). By the end of the season (in marked-sized fish), the contents of total saturates, monoenes, n-6 PUFAs and their ratios in the tissues of rainbow trout fed the two diets were not significantly different (Figure 4).

In the liver of both fish groups, the minimum level of essential linoleic (LA; 18:2n-6) and alpha-linolenic (ALA; 18:3n-3) fatty acids was noted in July (Figure 6). In fish muscle, the ALA relative abundance increased throughout the season, while LA reserves were relatively similar until a substantial loss in August (in the control) or September (in supplement-fed fish; Figure 6).

Two factors affecting the lipid composition of *O. mykiss* organs were discriminated with the factor load analysis of the lipid variables. The first one, primarily affecting TAGs, sterols, EPA, DHA, ArA, 16:1n-7/16:0 and 18:1n-9/18:0 indices, was fish size (weight), and the second one, affecting PS content, the ratio of PEA to PC, the ratio of SFA to PUFA and unsaturation index, was recognized to be the ambient temperature varying with the sampling date (Table 3). The lipid composition was not substantially affected by the ‘dietary supplement’ factor if estimating the throughout-season effect (Table 3 and Table 4).

## 4. Discussion

### 4.1. Summary of Growth Performance

In the present study, the dietary supplement significantly increased the survival rate and performance in rainbow trout, with no statistical difference in terms of relative growth rate and specific growth rate. However, by the day of slaughtering, the commercial weight of rainbow trout fed the supplemented diet significantly exceeded those of the controls. Similarly, the growth-accelerating activity of dihydroquercetin has not been precisely confirmed in trials on fish and non-fish species, varying from moderate to negligible [33,38,39], whereas the growth promotion effect of relative substances like quercetin or rutin has been well established in fish [40,41,42]. Although numerically higher values were observed in supplemented diet-fed fish, there was no significant difference in the feed conversion ratio. Despite this, higher fish body weight and total fattiness mean that fish fed the supplemented diet utilized dietary lipids more efficiently than fish fed a standard one. Our previous observations on one-year-aged rainbow trout [34], stressed by bacterial infection, indicated a better assimilation of dietary nutrients, like ALA 18:3n-3 and DHA 18:2n-6 fatty acids, because of feed supplementation. Dietary supplement appears to also be beneficial for fish survival and performance under environmental stressors like high temperature. Our findings support the data on promoting a variety of inherent mechanisms, such as disease resistance [32,33], inherent immunity and gut microbiota composition [32], allowing them to survive, withstand and recover from a particular stressor or the multiple stressors associated with intense fish cultivation.

### 4.2. Growth-Related Dynamics in Lipid Accumulation

Total body growth in fish is tightly related to skeletal muscle growth and associated with lipid and protein accumulation as energy reserves. Feed for reared fish is commonly formulated, considering that the higher the fat content in trout feed, the greater the accumulation of fat in the body of the fishwill be, while exogenic stressors like high water temperature impact feeding behaviour, limiting feed needs [43]. Since no sporadic stressors, such as bacterial or parasitic infections, perturbed fish welfare during the observation period, both groups exhibited the characteristics associated with physiological growth under natural variations in the environment. In size-matched individuals, either supplement-free or supplement-fed, tissue lipid compositions were equal at the start of the season and changed in a similar growth-dependent manner throughout the observation period, with substantial differences between the samples in July (temperature maximum) and September (temperature lowering). We found that tissue content of individual lipid classes primarily deriving from the diet correlated strongly with fish size and more readily responded to fish body growth than to exogenous stimuli (Table 2). The results showed a similar, mostly independent of a diet supplementation, rise in energy lipid stores via TAG, Chol and 16:1n-7 and 18:1n-9 monoene accumulation in fish organs during the growing season, indicating that they are readily assimilated from commercial feed enriched with TAGs and monoenes as the main components, contributing to fish growth. Hepatic and muscle PUFAs, mainly EPA, DHA and ArA, in an opposite manner, decreased with fish growth because of the relative abundance of the tissues with TAGs, containing mainly saturates and monoenes but few PUFAs. The observed seasonal decrease in PUFA tissue contents probably results from the PUFA-deficient diet, inhibition of PUFA biosynthesis by feed-derived MUFAs and intensive PUFA consumption for fish growth. By the end of the growing season, the supplement-fed fish had higher final weight, and their skeletal muscle and liver reserved more fats of various classes, principally TAGs and Chol. To conclude, a linear increase in fish growth parameters depending on the accumulation of reserve lipids and their fatty acids, primarily saturated and monoenoic, is mostly supplementindependent and may be perturbed by exogenous impacts, such as suboptimal water temperatures.

### 4.3. Temperature Response

In poikilothermic organisms like fish, ambient temperature affects the total lipid accumulation, which is most intensive at a temperature range optimal for growth (14–19 °C for rainbow trout) and induces modifications in lipid composition to maintain proper membrane permeability while the temperature decreases/increases [3,4,5]. Commonly, fish respond to hypo- and hyperthermia through the suppression of nutrient assimilation and lipid accumulation, and, at the cellular level, thermal response engages the percentage of PEA, PC and the SFA/PUFA ratio in membrane lipids. In our study, up to a three-fold water temperature increase (from 7.5 °C in May to 24.3 °C in July) substantially impacted the physiology of studied trout, including tissue lipid accumulation and their composition. The thermal response in fish was expectedly manifested by the increased ratio PEA to PC and total PUFA content at low temperatures (May samples) and a decrease in the ratio of PEA to PC and total PUFAs at temperature maximum. An observed throughout-season increase in phosphatidylserine, a minor component of the membrane lipid bilayer, possibly indicates its specific contribution to cell cycle signalling and membrane ion permeability in intensely growing fish; in addition, in summer, it was less consumed for the enzymatic conversion to PEA than at low temperatures. In accordance with a contribution of fatty acids to the maintenance of functional membrane fluidity, their saturation index SFA/PUFA was lower under low water temperatures (May samples), and, oppositely, saturated fatty acids prevailed in total lipids under high temperatures (July samples). Tissue PUFA abundance is controlled in multiple ways, from their consumption with feed, synthesis de novo and mutual conversion to beta-oxidation in the electron transport chain [44]. In addition, freshwater fish are capable of producing PUFAs, particularly EPA and DHA from linolenic acid and ArA from linoleic acid, through the activities of elongase and desaturase [45,46]. Those gene expressions could be downregulated in response to external triggers like ambient temperature peak. In the studied fish, PUFA synthesis pathways could also be inhibited by both the excess in MUFA assimilated from high-fat commercial feed and a substrate depletion revealed by the substantial loss in LA 18:2n-6 and ALA 18:3n-3 fatty acids known to be n-3 and n-6 PUFA precursors. Detected exhaustion in the EPA pool, a source for proinflammatory eicosanoids, possibly indicates inflammation to be a part of the high-temperature response in fish [47,48]. 

Despite the temperature responses in the studied fish being in accordance with a well-known scenario, our work is the first study addressing the effect of a bioactive natural supplements, a dihydroquercetin and arabinogalactan mix, on seasonal temperature-dependent lipid and fatty acid patterns in healthy rainbow trout. With an equal rise in muscle lipid stores, both in control and experimental fish, in the periods favourable for growth, diet supplementation partially mitigates the suppression of tissue lipid accumulation induced by an ambient temperature peak (20–25 °C, late July). The high-temperature response was less pronounced in fish fed the supplemented diet, indicating a higher physiological tolerance to the temperature rise. The lipid metabolism of fish appears to respond in a similar way to different stressors. For example, the lipid profile observed in our study in rainbow trout affected by a seasonal temperature rise was similar to that of trout fed an alimentary inadequate diet [49], with a lower level of n-3 PUFAs, namely EPA 20:5n-3 and DHA 22:6n-3, and higher amounts of n-6 PUFAs, including γ-linolenic 18:3n-6 and linoleic 18:2n-6, in fish tissues. Similar long-persisting lipid abnormalities were shown in diseased one-year-old rainbow trout [5] when a bacterial mixinfection led to the substantial suppression of physiological accumulation of TAGs and PLs, a shift in membrane lipid composition, including sterols, PEA and vaccenic 18:1n-7 fatty acid, as well as the prevalence of n-6 on n-3 PUFAs. The described effects of the environmental and biotic (infectious) stressors on trout lipid composition were shown to also be ameliorated by the dihydroquercetin-supplemented diet [5].

### 4.4. Fillet Lipid and Fatty Acid Composition

Our findings support the observations that the physiological growth process and environmental variables both affect lipid and fatty acid composition in trout white muscles. The summer rise in water temperature was above the optimal for stenobiont species, as rainbow trout impact fish stock and moderately deteriorate nutritional value and consumer attributes of fish fillet through modifying the lipid composition. Market-sized O. mykiss (slaughtered at the end of the season) seems to be less balanced and contains lower levels of essential ALA, ArA and PUFAs but not monoenes compared with the same fish sampled at the beginning of the season (in May) while more balanced if compared with the same fish sampled in mid-summer, apparently due to changes in lipids induced by high temperatures. Progressive exhaustion of health-promoting substances, like PUFAs, in the fillet of trout during a season may derive from both the increase in fish total fat (particularly TAG) accumulation related to fish growth and the PUFA-deficient composition of commercial feed for fish. In the current study, a supplemented diet resulted in moderate but significant changes in the lipid composition of the fish fillet. Firstly, in the period unfavourable for fish growth (in mid-summer), supplement-fed fish muscles prevailed in the ratio of n-3 to n-6 fatty acids and contained less cholesterol. Secondly, by the time of slaughtering, the commercial-size trout grown on supplemented feed turned out to have a higher nutritional and energetic value, due to higher total fattiness, TAG and PL contents, if compared with standard-fed fish. Although the tested supplement was found to be moderately beneficial in improving lipid and fatty acid composition in stressed fish, taken together with the data on survival rate, our observations suggest the dietary supplement’s effectiveness in increasing the ability of reared fish to cope with environmental stressors as well as in increasing aquaculture sustainability. Our data assume that other ways to enrich fish products with health-promoting substances should also be developed, for instance, through the direct inclusion of lipid molecules beneficially affecting human health, such as krill meat rich in n-3 PUFA [50] and plant oils rich in ALA and LA, the precursors in n-3 and n-6 PUFA biosynthesis [51,52], in fish feed recipes. The cardioprotective lipid profile of the fillet could be maintained by providing fish with a high n-3 PUFA finishing diet prior to the slaughtering [53].

## 5. Conclusions

Our results extend the knowledge on the effects of dietary supplementation with the bioactive compounds of plant origin, such as an antioxidant dihydroquercetin and a polysaccharide arabinogalactan, on the physiological and biochemical variables in farmed rainbow trout. While previous data [5,34] were obtained from fish subjected to a more complex influence, including sporadic infectious disease and maladaptation, our study is the first to address the role of dietary supplementation on lipid composition in healthy individuals growing under environmental variables. In undisturbed fish, dietary dihydroquercetin and arabinogalactan do not interfere with the growth physiology involving the assimilation of feed-derived lipid components and their accumulation in fish tissues, and the mode of temperature adaptation via the shift in membrane lipid composition was conserved in poikilotherms. Our work provides promising results on feed supplementation to mitigate the high-temperature response in rainbow trout and to improve the lipid and fatty acid composition of fillets produced from environmentally stressed fish. As the stresses associated with intensive fish production are unavoidable and unpredictable, the versatile welfare and tissue composition-related effects of the natural supplement could be used for prophylactic purposes. In addition, the dietary supplement studied is available, inexpensive, has no negative effects on fish physiology and is environmentally safe for the natural waters used for fish production. The results obtained are relevant to both industry and fundamental science, considering the impending increase in water temperatures, the continuous growth of aquaculture, the resources devoted to aquafeed and the much-needed preventive strategies in fish nutrition to achieve increased welfare and the adaptive potential of reared fish.

## Figures and Tables

**Figure 1 animals-14-00094-f001:**
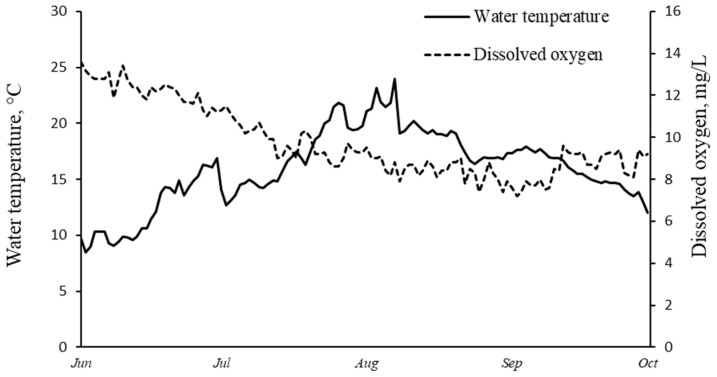
Water temperature (°C) and dissolved oxygen content (mg/L) in the surface water within the cages during the growing season.

**Figure 2 animals-14-00094-f002:**
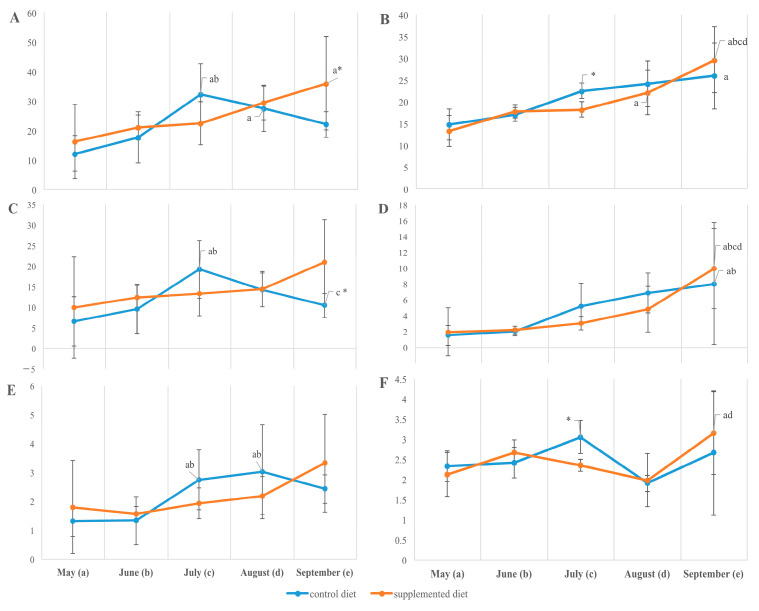
Total lipids (**A**,**B**), triacylglycerols (**C**,**D**), and cholesterol (**E**,**F**) (% dry weight) in the muscle (**A**,**C**,**E**) and liver (**B**,**D**,**F**) of *O. mykiss* fed control or supplemented diet. a, b, c, d, significant differences when compared with samples collected in May (a), June (b), July (c), and August (d); * significant differences when comparing both diets (*p* < 0.05).

**Figure 3 animals-14-00094-f003:**
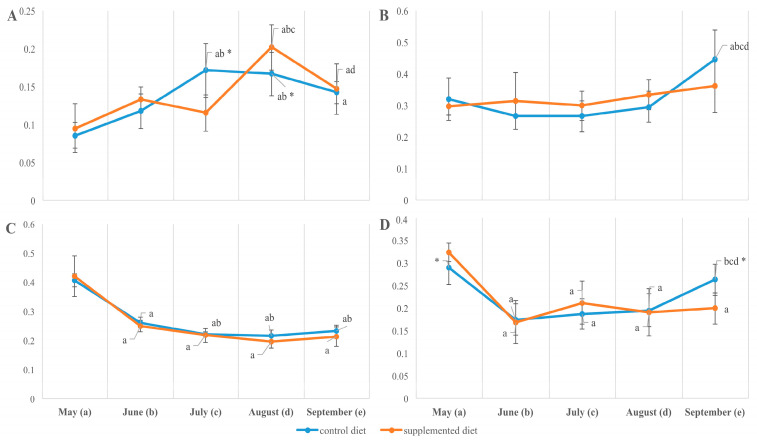
Phosphatidylserine (% dry weight, (**A**,**B**)) and phosphatidylethanolamine/phosphatidylcholine ratio (**C**,**D**) in muscle (**A**,**C**) and liver (**B**,**D**) total lipids in *O. mykiss* fed control or supplemented diet. a, b, c, d—significant differences when compared with samples collected in May (a), June (b), July (c), and August (d); * significant differences when comparing both diets (*p* < 0.05).

**Figure 4 animals-14-00094-f004:**
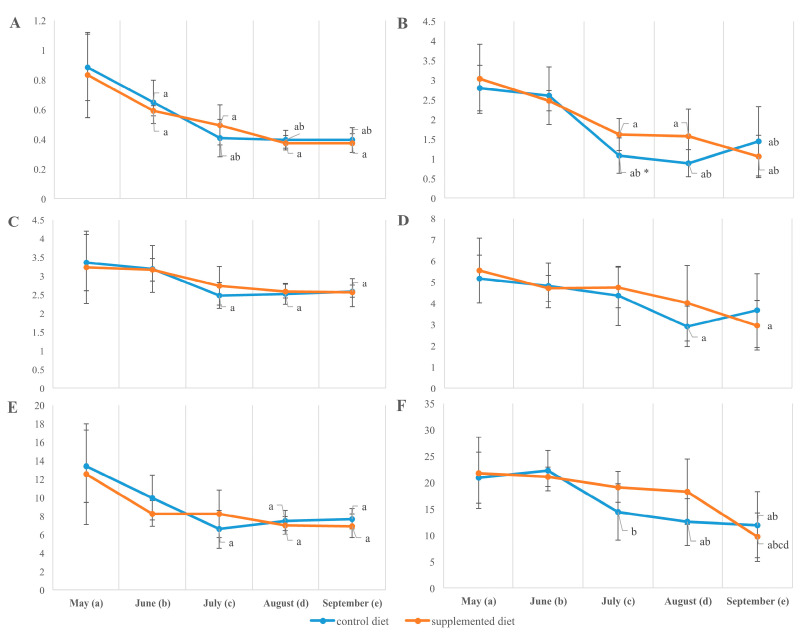
Arachidonic 20:4n-6 (**A**,**B**), eicosapentaenoic 20:5n-3 (**C**,**D**), and docosahexaenoic 22:6n-3 (**E**,**F**) fatty acid contents (% total fatty acids) of muscle (**A**,**C**,**E**) and liver (**B**,**D**,**F**) total lipids in *O. mykiss* fed control or supplemented diet. a, b, c, d—significant differences when compared with samples collected in May (a), June (b), July (c), and August (d); * significant differences when comparing both diets (*p* < 0.05).

**Figure 5 animals-14-00094-f005:**
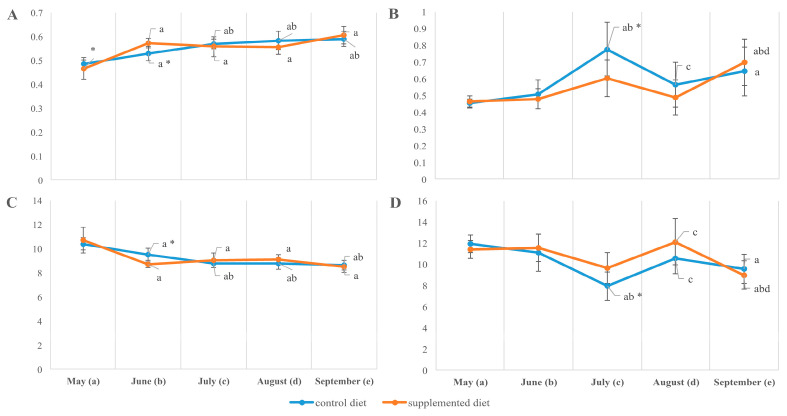
Saturated to polyunsaturated fatty acid ratio (**A**,**B**) and unsaturation index (**C**,**D**) of muscle (**A**,**C**) and liver (**B**,**D**) total lipids in *O. mykiss* fed control or supplemented diet. a, b, c, d—significant differences when compared with samples collected in May (a), June (b), July (c), and August (d); *significant differences when comparing both diets (*p* < 0.05).

**Figure 6 animals-14-00094-f006:**
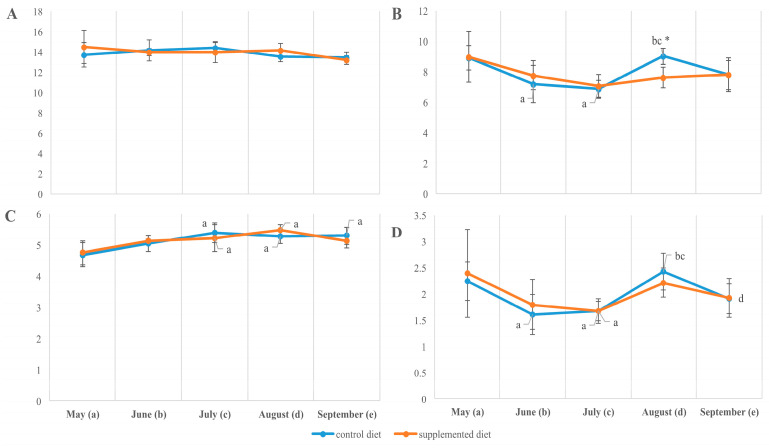
Linoleic 18:2n-6 (**A**,**B**) and alpha-linolenic 18:3n-3 fatty acids (**C**,**D**) in muscle (**A**,**C**) and liver (**B**,**D**) total lipids in *O. mykiss* fed control or supplemented diet. a, b, c, d—significant differences when compared with samples collected in May (a), June (b), July (c), and August (d); * significant differences when comparing both diets (*p* < 0.05).

**Table 1 animals-14-00094-t001:** Lipids and fatty acid composition of a basal diet.

Pellet Size	6 mm	8 mm
Total lipids, % dry weight	23.28 ± 0.37	24.03 ± 0.53
Main lipid classes, % total lipid dry weight		
Phospholipids	0.19 ± 0.17	0.16 ± 0.24
Triacylglycerols	18.68 ± 0.45	18.55 ± 0.94
Cholesterols	4.60 ± 0.40	5.32 ± 0.63
Fatty acid compositions, % sum fatty acids		
14:0	2.48 ± 0.04	2.35 ± 0.04
16:0	9.65 ± 0.15	8.73 ± 0.24
16:1n-7	2.22 ± 0.04	2.15 ± 0.02
18:0	2.61 ± 0.13	2.44 ± 0.23
18:1n-9cis	41.15 ± 0.18	39.72 ± 0.18
18:2n-6cis	16.15 ± 0.09	16.15 ± 0.10
18:3n-3	8.49 ± 0.06	8.52 ± 0.05
20:1n-9	2.16 ± 0.12	3.38 ± 0.06
20:5n-3	4.99 ± 0.08	4.25 ± 0.03
22:1n-9	2.40 ± 0.44	3.78 ± 0.41
22:6n-3	4.24 ± 0.08	5.63 ± 0.06

**Table 2 animals-14-00094-t002:** Growth performance and mortality of rainbow trout fed standard or supplemented diet for four-month trial. * significant differences when comparing both diets (*p* < 0.05).

Variables	Standard Diet	Supplemented Diet
Mortality (%)	3.65 ± 0.33	2.88 ± 0.32 *
Initial weight (g)	733.0 ± 58.3	770.1 ± 56.5
Final weight (g)	1947.3 ± 314.7	2289.3 ± 120.3
Relative growth rate (%)	286.4 ± 42.8	313.2 ± 67.7
Specific growth rate (% day^−1^)	1.47 ± 0.27	1.60 ± 0.46
Feed conversion ratio	1.27 ± 0.08	1.31 ± 0.06

**Table 3 animals-14-00094-t003:** Factor loadings (varimax raw) (bold marked loadings are >0.7).

Variables	Muscle		Liver	
	Factor 1	Factor 2	Factor 1	Factor 2
Total lipids	0.42	**0.89**	**0.96**	0.21
Phospholipids, PLs	**0.90**	0.42	**0.94**	0.01
Triacylglycerols, TAGs	0.20	**0.93**	**0.87**	0.29
Cholesterols, Chol	0.27	**0.85**	**0.70**	0.20
Phosphatidylserine, PS	**0.89**	0.31	0.35	**0.80**
Phosphatidylethanolamine, PEA	**0.90**	0.01	0.22	**0.92**
Phosphatidylcholine, PC	**0.86**	0.49	**0.92**	−0.26
PEA/PC ratio	−0.29	−0.61	−0.52	**0.83**
Expl.Var	3.50	3.26	4.31	2.40
Prp.Totl	0.44	0.41	0.54	0.30
Fatty acid composition				
	Factor 1	Factor 2	Factor 1	Factor 2
16:0, palmitic	0.44	**−0.84**	0.47	**−0.71**
18:0, stearic	0.38	−0.66	0.44	**−0.72**
Total saturate, SFA	0.21	**−0.92**	0.52	**−0.81**
16:1n-7, palmitoleic	**−0.91**	−0.01	**−0.88**	−0.32
18:1n-9, oleic	**−0.85**	0.50	**−0.94**	0.27
18:1n-7, vaccenic	−0.37	0.29	−0.67	0.29
Total monoenes, MUFA	**−0.91**	0.38	**−0.97**	0.21
18:2n-6, linoleic, LA	−0.13	**0.93**	−0.17	**0.82**
18:3n-6, γ-linolenic	0.39	0.22	0.04	0.02
20:4n-6, arachidonic, ArA	**0.92**	−0.25	**0.92**	0.04
Total n-6 PUFA	0.20	**0.88**	0.57	**0.71**
18:3n-3, α-linolenic, ALA	−0.69	0.43	−0.13	**0.74**
20:5n-3, eicosapentaenoic, EPA	**0.82**	−0.40	**0.90**	−0.22
22:6n-3, docosahexaenoic, DHA	**0.91**	−0.38	**0.98**	−0.12
Total n-3 PUFA	**0.92**	−0.38	**0.99**	−0.08
Total PUFA	**0.98**	−0.16	**0.99**	0.06
n-3/n-6 PUFA ratio	**0.81**	−0.55	**0.87**	−0.38
SFA/PUFA ratio	**−0.88**	−0.44	**−0.70**	−0.66
16:1n-7/16:0 ratio	**−0.84**	0.38	**−0.94**	−0.04
18:1n-9/18:0 ratio	−0.68	0.66	**−0.77**	0.49
18:1n-7/16:1n-7 ratio	**0.79**	0.10	**0.80**	0.37
Unsaturation index	**0.92**	0.26	0.51	**0.78**
Expl.Var	11.78	6.03	12.30	5.35
Prp.Totl	0.54	0.27	0.56	0.24

**Table 4 animals-14-00094-t004:** Correlations of muscle and liver lipids and fatty acids with the ‘sampling date’, ‘fish weight’, and ‘dietary supplement’ factors (bold are marked reliable correlations).

Variables	Muscle			Liver		
	Sampling Date	Fish Weight	Dietary Supplement	Sampling Date	Fish Weight	Dietary Supplement
Total lipids	**0.50**	**0.53**	−0.13	**0.65**	**0.68**	0.03
Phospholipids, PLs	**0.60**	**0.57**	−0.04	**0.62**	**0.63**	0.06
Triacylglycerols, TAGs	**0.34**	**0.40**	−0.15	**0.60**	**0.64**	0.02
Cholesterols, Chol	**0.45**	**0.48**	−0.02	0.16	**0.25**	0.02
Phosphatidylserine, PS	**0.56**	**0.52**	−0.01	**0.41**	**0.36**	−0.01
Phosphatidylethanolamine, PEA	0.19	0.20	0.05	0.20	0.18	0.10
Phosphatidylcholine, PC	**0.64**	**0.61**	−0.05	**0.56**	**0.58**	0.05
PEA/PC ratio	**−0.75**	**−0.68**	0.04	**−0.31**	**−0.34**	0.00
16:0, palmitic	−0.19	−0.16	0.14	**−0.37**	**−0.44**	0.01
18:0, stearic	**−0.23**	−0.18	0.11	−0.15	−0.16	0.02
Total saturate, SFA	−0.03	−0.01	0.10	**−0.30**	**−0.36**	0.02
16:1n-7, palmitoleic	**0.59**	**0.64**	−0.08	**0.61**	**0.64**	0.08
18:1n-9, oleic	**0.51**	**0.51**	−0.04	**0.53**	**0.57**	0.10
18:1n-7, vaccenic	**0.29**	**0.27**	−0.02	**0.76**	**0.74**	−0.01
Total monoenes, MUFA	**0.63**	**0.61**	−0.06	**0.63**	**0.67**	0.08
18:2n-6, linoleic, LA	**−0.27**	**−0.25**	−0.06	−0.18	−0.14	0.03
18:3n-6, γ-linolenic	**−0.35**	**−0.42**	0.16	−0.19	−0.21	0.18
20:4n-6, arachidonic, ArA	**−0.74**	**−0.70**	0.03	**−0.69**	**−0.69**	−0.08
Total n-6 PUFA	**−0.57**	**−0.54**	−0.04	**−0.61**	**−0.58**	−0.07
18:3n-3, α-linolenic, ALA	**0.50**	**0.52**	−0.01	−0.07	−0.02	−0.05
20:5n-3, eicosapentaenoic, EPA	**−0.49**	**−0.45**	−0.01	**−0.52**	**−0.55**	−0.06
22:6n-3, docosahexaenoic, DHA	**−0.56**	**−0.57**	0.06	**−0.61**	**−0.64**	−0.10
Total n-3 PUFA	**−0.56**	**−0.55**	0.05	**−0.61**	**−0.64**	−0.10
Total PUFA	**−0.71**	**−0.70**	0.04	**−0.66**	**−0.68**	−0.11
n-3/n-6 PUFA ratio	**−0.39**	**−0.39**	0.06	**−0.44**	**−0.48**	−0.09
SFA/PUFA ratio	**0.73**	**0.73**	0.00	**0.45**	**0.45**	0.13
16:1n-7/16:0 ratio	**0.49**	**0.53**	−0.12	**0.66**	**0.70**	0.04
18:1n-9/18:0 ratio	**0.40**	**0.38**	−0.08	**0.42**	**0.45**	0.08
18:1n-7/16:1n-7 ratio	**−0.47**	**−0.52**	0.06	**−0.52**	**−0.51**	−0.16
Unsaturation index	**−0.67**	**−0.69**	−0.02	**−0.37**	**−0.32**	−0.12

## Data Availability

All data are presented in the paper.

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
