# Peer review of "Tissue Lipid Profiles of Rainbow Trout, Oncorhynchus mykiss, Cultivated under Environmental Variables on a Diet Supplemented with Dihydroquercetin and Arabinogalactan"

_animals, 2023, doi:10.3390/ani14010094_

Round 1

Reviewer 1 Report

Comments and Suggestions for Authors

This study aimed to find the effects of dietary Dihydroquercetin and Arabinogalactan supplementation on growth performance, survival, and fillet fatty acid composition of trout, during one year, including warm months. There are methodological and structural issues in this study that make it unsuitable for publication.

 The authors had only two cages per treatment, meaning only 4 numeric for the statistical analysis at each time. This is a low sample size. Also, the authors used one way ANOVA for data analysis that is not correct. As the fish were sampled from the same cage at different times, repeated measure two-way ANOVA must be used. It is not clear how the Varimax coefficients were judged as reliable/no-reliable correlation. Normally, coefficients higher than 0.5 or lower than -0.5 should be considered as reliable correlation.

 There is no basis for choosing the concentrations of Dihydroquercetin and Arabinogalactan. The authors cited a certificate, but I could not find it. In the supplier website, there is no data regarding the use of these products in aquaculture, but livestock, bees, pharmaceuticals .... Also, it is not how the authors decided to feed the supplemented diet to fish for 2 weeks, followed by two-weeks of feeding by the control diet! This may explain why there were no noticeable differences/correlations in the measured parameters between the treatments.

 The data show that either the fish were resistant to high temperature (probably due to selection) or the peak temperature was not stressful. Mortality in both groups were very low (less than 4%), which is not a characteristic of chronic stress. Also, there are no noticeable changes in fatty acid composition between the treatments.

 Structural issues:

Simple summary: this section must emphasize the possible application of the study results in the field.

Abstract: It is general. No details of the experimental methods and conditions and results were provided.

Author Response

The authors are deeply grateful to the reviewers for their careful reading of the manuscript and for their valuable comments and suggestions. The title has been revised as the reviewer pointed out and now read as “Tissue Lipid Profiles of Rainbow Trout, Oncorhynchus mykiss, Cultivated under Environmental Variables on a Diet Supplemented with Dihydroquercetin and Arabinogalactan”. We have carefully revised the manuscript over based on the comments of four reviewers to improve the readability and scientific soundness of the manuscript. Our responses are given in attached file. 

Reviewer 2 Report

Comments and Suggestions for Authors

This manuscript entitled " Dietary Dihydroquercetin and Arabinogalactan Mitigate Heat-Induced Changes in Tissue Lipid Profiles of Reared Rainbow Trout, Oncorhynchus mykiss" by Fokina et al. is an interesting manuscript focused in evaluating the effect of mix supplements on the changes in lipid profile of rainbow trout under high temperature induction under culture conditions.

The paper presents a well sampling methods and analytical methodology, with many results in the lipid and fatty acid profile changes during stress and regular culture conditions. At same time, clear differences in results are shown by the inclusion of Dihydroquercetin and Arabinogalactan. The presentation of results is clear, and the discussion is well written. However, there are some comments that should be addressed to enrich the quality of the manuscript.

Major

In the introduction, there is no mention of the use of these or other flavonoids or polysaccharides in fishes, or the use of the flavonoids used in the present work in other animals, as well as their effects. In the previous article (22), some backgrounds are stated that are not presented in this manuscript.

One supplement is polyphenol and the other polysaccharide, the action at metabolic level is different and is expected that a Synergy could occur and maybe potentialize the effects. However, a lack in information about the individual action of each supplement is clear. Nothing justifies the proportions used (line 85-86)? Background information is needed.

In the introduction, the beneficial effects are mentioned in line 70-71, however, it is not clear how these supplements (polyphenols and polysaccharides) would generate a differential effect on the fatty acid profile? Do these polyphenols improve the microbiota? Does the difference in accumulation occur at the digestive level as well? Are they absorbed? Do they improve the antioxidant system? There is no mention of the method of action, either in this section or in discussion. The entire discussion focuses on the accumulation and utilization of these reserves, but never mentions how these supplements activate metabolic pathways and at what level to generate the lipid difference that the write-up focuses so much on.

Minors

Line 19. What does the word normal refer to? Within what parameters? The normal word is confusing.

Line 20 and 36. What does a favorable period refer to?

Line 25. Lipid composition partially alleviated?

Line 353 and 355. Season? Which one?

Line 451. Do not interfere with their physiology?? This contradicts all the results, which show differential lipid accumulation and lipid spending by the supplements.

Author Response

The authors are deeply grateful to the reviewers for their careful reading of the manuscript and for their valuable comments and suggestions. The title has been revised as the Reviewer pointed out and now read as “Tissue Lipid Profiles of Rainbow Trout, Oncorhynchus mykiss, Cultivated under Environmental Variables on a Diet Supplemented with Dihydroquercetin and Arabinogalactan”. We have carefully revised the manuscript over based on the comments of four reviewers to improve the readability and scientific soundness of the manuscript. Our responses are given in attached file.

Reviewer 3 Report

Comments and Suggestions for Authors

Rainbow Trout, Oncorhynchus mykiss is a very economically valuable fish species. The study is relevant and very interesting and is of great importance to the aquaculture industry. However, there are still some areas where minor modifications are needed.

Question 1 What are the commercial selling prices for dihydroquercetin and arabinogalactan?

Question 2 Introduction are too brief and need to be added appropriately.

Question 3 What is the basis for the additive quantities of dihydroquercetin and arabinogalactan?

Question 4 Why is it important to change feed pellets of different diameters at different feeding periods? Is this consistent with actual production feed pellets?

Comments on the Quality of English Language

There is still significant space for improvement in English writing. It is recommended to carefully re-organize the language and reduce loopholes.

Author Response

(The authors gave the same response as above.)

Reviewer 4 Report

Comments and Suggestions for Authors

This manuscript examines the effects of dietary supplementation with dihydroquercetin and arabinogalactan on lipid accumulation and fatty acid composition in rainbow trout reared under environmental variables. The research is interesting and the results suggest some benefits of supplementation in mitigating high temperature stress effects. However, there are a couple issues that should be addressed:

1) The title implies that there were heat stress experiments performed, but the methods do not describe any temperature manipulation experiments. The fish were reared under natural seasonal temperature fluctuations in lake cages. To avoid overstating, I suggest revising the title to more accurately reflect the experimental conditions, e.g. "...reared under environmental variables" instead of "...heat-induced changes."

2) Along the same lines, statements in the abstract about "heat-induced depression of lipid accumulation" and paragraphs in the discussion about "high-temperature response" and "heat stress response" are not well supported since no intentional heat stress was imposed. I recommend toning down these parts to focus instead on the natural temperature effects observed.

3) Some statements connecting effects on lipid composition to impacts on fish welfare and fillet quality should be qualified, as the implications were not directly assessed. For example in the conclusions the statements "accelerating fish tolerance to environmental stressors" and "improving fillet quality attributes" require more substantiation.

Overall the article provides helpful new data, but revisions are needed to ensure the title, abstract, and interpretations accurately reflect the environmenal rearing study design. Adding supporting temperataure manipulation experiments in the future would strengthen the heat stress aspects.

Author Response

(The authors gave the same response as above.)

Round 2

Reviewer 1 Report

Comments and Suggestions for Authors

Although the authors have made corrections in some cases that help to improve the quality of the manuscript, the answers to the questions related to the design and implementation of the project were not convincing, and unfortunately, I cannot recommend its publication.

Reviewer 2 Report

Comments and Suggestions for Authors

The authors addressed all comments, improved the introduction. The manuscript is well written, and the results are clear and positive. 

One detail to consider for future work is not to test concentrations and mixtures with respect to a supplier or a developed brand name product. The concentrations are species specific and it is always necessary to evaluate the additives separately before mixing them at their previously determined optimums. On this occasion their inclusion was positive, but we still do not know if they really synergize, we only know the final results.

Reviewer 3 Report

Comments and Suggestions for Authors

In my opinion, the authors have answered all questions last round, and the paper can be accepted in the current version.

Comments on the Quality of English Language

In my opinion, the authors have corrected most of the grammatical problem last round, and the paper can be accepted in the current version.

Reviewer 4 Report

Comments and Suggestions for Authors

Good